# Prevalence of Chronic Kidney Disease in Cuttack District of Odisha, India

**DOI:** 10.3390/ijerph17020456

**Published:** 2020-01-10

**Authors:** Nisith Kumar Mohanty, Krushna Chandra Sahoo, Sanghamitra Pati, Asish K. Sahu, Reena Mohanty

**Affiliations:** 1Utkal Kidney Foundation, Apollo Hospitals, Bhubaneswar 751005, Odisha, India; nkmty2002@yahoo.co.in (N.K.M.); dipakaten@gmail.com (A.K.S.); 2Regional Medical Research Centre, Indian Council of Medical Research, Bhubaneswar 751023, Odisha, India; drsanghamitra12@gmail.com; 3Capital Hospital, Bhubaneswar 751001, Odisha, India; reenamohanty80@gmail.com

**Keywords:** CKD prevalence, Odisha, Cuttack

## Abstract

Chronic kidney disease is one of the major health challenges in India. Cuttack district of the Odisha state of India is regarded as a hotspot for chronic kidney disease (CKD). However, there is limited information on true prevalence. This study estimates the prevalence of CKD in the Narsinghpur block of Cuttack district, Odisha. A cross-sectional study was conducted among population members aged 20–60 years. Using a multi-stage cluster sampling. 24 villages were randomly selected for mass screening for CKD. Blood samples were collected and glomerulus filtration rates were calculated. It was found that among the 2978 people screened, 14.3% were diagnosed with CKD and 10.8% were diagnosed with CKD without either diabetes or hypertension. In one-third of the sampled villages, about 20% population was diagnosed with CKD. The prevalence was higher among males (57%), in the population below 50 years of age (54%), lower socioeconomic groups (70%), and agricultural occupational groups (48%). Groundwater tube wells (49%) and wells (41%) were the main drinking water sources for CKD patients. This study highlights the need for detection of unknown etiologies of CKD and public health interventions for the prevention of CKD in India.

## 1. Introduction

Chronic kidney disease (CKD) is defined as kidney damage with a glomerular filtration rate (GFR) < 60 mL/min/1.73 m^2^ for three months or more, irrespective of the cause. Kidney damage in many kidney diseases can be ascertained by the presence of albuminuria, defined as an albumin–creatinine ratio >30 mg/g in two of three spot urine specimens [1]. It affects the kidney structure, reduces the glomerular filtration rate, and increases albumin excretion in urine [1,2]. It is one of the rising public health issues worldwide [3,4,5]. The global prevalence of CKD is 8%–16% and is the twelfth most common reason for mortality [3,6,7]. The consequences of CKD are increased cardiovascular disease, premature-mortality, and degraded quality of life [2]. Globally, 80% of CKD related deaths occur in developing countries [8,9,10]. However, there is a lack of kidney registries in many developing countries and it is a challenge to estimate the true prevalence of CKD in these countries [8,9,10]. In developing countries, the majority of patients died because of the unaffordability of high cost treatments [5,11,12]. 

CKD is one of the foremost public health concerns in Southeast Asia [12,13]. However, the true incidence and prevalence in these regions are lacking. Most of the affected population are younger and in their productive years of life [12,14]. Similarly, in India, CKD is one of the common non-communicable diseases and results in high morbidity, mortality, and economic burden [3,15]. In India, the number of CKD related mortalities was about 5.2 million in 2008 and may rise to 7.63 million by 2020 [16]. Hot-spots of CKD have been reported in a few states—India-Puducherry, Andhra Pradesh, Maharashtra, and Odisha [15]. However, in India, accurate prevalence and incidence data of CKD is lacking [8,17,18] as there are hardly any community-based studies [15]. 

In Odisha, Cuttack district is regarded as a hotspot for CKD. According to hospital records, more than 2000 cases have been identified during the last five years, of which nearly 25% died. However, all reported cases were based on hospital records. There is limited information on the true prevalence of CKD in Odisha. Therefore, this study estimated the prevalence of CKD in Narsinghpur, Cuttack, Odisha, India. 

## 2. Materials and Methods

### 2.1. Study Design and Settings

Cross-sectional epidemiological research was conducted to estimate the actual prevalence of CKD in the Narsinghpur block of Cuttack, Odisha. There are 25 administrative blocks in Cuttack district with a total population of 2.62 million. The Narsinghpur block has 157 villages and a population of about 98,000, with around 58,000 people between 20–60 years of age. There was a representative sample of the general population in this study. The study included male and female members of the population in the 20–60 years of age group. 

### 2.2. Sample Size, Sampling, and Study Variables

The required sample size for this study was calculated using Epi Info software. According to Varma in India, the prevalence of CKD is 6.3% among the younger population, even those with low prevalence of diabetes [9]. The required sample size was 2268 with an estimated proportion of 6.3%, a confidence level 95%, and a desired precision of 0.01. 

The sample was collected using a multi-stage cluster method. The universe of the study was the population of 20–60-year-olds in the Narsinghpur block (*n* = 58,574), distributed across 157 villages. In order to achieve the required number of samples, 24 villages were randomly chosen from 157 villages. The socio-demographic profile, occupational history of the participants, information on CKD history, and other chronic illnesses were collected. The weight and height of the participants were measured for the calculation of Body Mass Index (BMI). 

### 2.3. Collection and Analysis of a Clinical Sample

The clinical measurements included blood pressure, random blood sugar, urine protein and sugar, and serum creatinine. Hypertension is defined as a systolic BP ≥ 140 mmHg or diastolic BP ≥ 90 mmHg. A random blood glucose level of 200 mg/dL or more, with the presence of symptoms such as increased urination as well as thirst and unexplained weight loss, were considered as evidence of the individual having diabetes. Urine was collected in a transparent plastic container and tested immediately using a dipstick to test protein and sugar (Catalog No. 2161; MultiStix 10SG, Urine Dipstick, 100 strips per package, Bayer). 

To process the test, 7.5 mL of blood was collected by trained professionals. The samples were processed and the biochemical test was performed using standard protocol at GenX Diagnostics, Bhubaneswar, which is certified by the National Accreditation Board for Testing and Calibration Laboratories. The creatinine was tested using the enzymatic method and glucose by the hexokinase method. The Glomerular Filtration Rate (GFR) was calculated with the help of the Modification of Diet in Renal Disease formula, Primary Care Informatics, London, United Kingdom. The CKD stages were estimated using the National Kidney Foundation Kidney Disease Outcomes Quality Initiative guidelines [19].

### 2.4. Statistical Analysis

The data were entered into Microsoft Excel and then transferred to Stata 10.1 (Stata Corp. College Station, TX, USA) software for statistical analysis. Categorical variables were expressed in terms of frequencies (*n*) and percentages (%). The quantitative variables were expressed through mean and standard deviation (*SD*). The characteristics of the screening population and prevalence of CKD were presented in the table. The prevalence of various stages of CKD was presented using a bar diagram. Differences in prevalence were assessed by a chi-square test and *p* < 0.05 was considered statistically significant.

### 2.5. Ethical Considerations

The Institutional Review Board of Apollo Hospitals, Bhubaneswar approved this study. Written consent was obtained from all participants for the collection of information and clinical samples. 

## 3. Results

A total of 2616 households in 24 villages were surveyed. Out of 11,481 individuals, 6581 were in the age group of 20–60 years. Of these, 1970 were staying outside of the village and 478 were absent during the survey. Hence, the target population for screening was 4133 individuals. However 1155 were unwilling to participate in this study and a total 2978 individuals participated in this study. 

Among 2978 individuals, 349 (11.7%) had hypertension, 169 (5.7%) had diabetes, 459 (15.4%) were either diabetic or had hypertension, and only 59 (1.98%) had both diabetes and hypertension.

Out of the screened population of 2978, 426 (14.3%) were diagnosed with CKD. Among them, a total of 323 (10.8%) were diagnosed with CKD without diabetes or hypertension and 103 (3.5%) had CKD with either diabetes or hypertension. Only 18 individuals had CKD with both diabetes and hypertension. The *mean* (± *S.D*) *value*
*of* serum creatinine in CKD patients with either diabetes or hypertension (*n* = 103) was observed to be 157.06 ± 98.41, whereas in CKD patients without diabetes or hypertension (*n* = 323) it was 171.67 ± 135.35. Similarly, the *mean* (± *S.D*) *value of* GFR in CKD patients with either diabetes or hypertension (*n* = 103) was observed to be 50.7 ± 25.27, whereas in CKD patients without diabetes or hypertension (*n* = 323) it was 50.81 ± 27.2. In Table 1 the characteristics of the screening population and prevalence of CKD are presented.

The results show that the prevalence of CKD is higher among females (53%) in comparison to males (47%). However, the occurrence of CKD in patients without diabetes or hypertension is higher among males (57%) than females (43%). Among the patients having CKD without diabetes or hypertension, more than half (54%) of them were below 50 years of age. However, in cases of CKD with either diabetes or hypertension, 57% were above 50 years of age. 

It was found that among the patients having CKD without diabetes or hypertension, 46% belonged to other backward classes and 28% to scheduled castes. A majority of them belonged to sub-caste chase/sudra (18%), khandayat (19%), and kaibarta (15%). Around 26% had an agricultural occupation and 22% were agricultural laborers. Around 70% belonged to the lower socioeconomic group. The main drinking water sources were groundwater, such as from tube-wells (49%) and wells (41%). Among patients, 76% never consumed alcohol. In the villages studied, one-third of the villages and about 20% of the population were diagnosed with CKD without diabetes or hypertension. 

In Figure 1 the prevalence of various stages of CKD with either diabetes or hypertension and without diabetes or hypertension is presented. It was found that among the patients who had CKD without diabetes or hypertension, the prevalence was stages I (22.6%), III (54.2%), IV (16.7%), and V (6.5%). 

## 4. Discussion

In India, there is a paucity of data in the national registry on the incidence and prevalence of CKD [8,17,18]. Hence, the accurate burden of CKD in India is lacking. The approximate prevalence is around 800 per million population and the incidence of end-stage renal failure cases is about 200 per million population [20]. In our study total, 14.3% were diagnosed CKD and the prevalence of CKD without diabetes or hypertension was 10.8%. In India, a previous community-based screening showed about 17.2% of CKD cases [21]. A study conducted among 52,273 adults showed rates of CKD diagnosis in South India (36%), North India (28%), West India (26%), and East India (11%) [17]. 

The prevalence of various stages of CKD in our study was stages I (22.6%), III (54.2%), IV (16.7%), and V (6.5%). However, a study conducted among Indian central government employees over 18 years of age showed that about 6.6% had stage I, 5.4% stage II, and 3% stage III CKD [18]. Similarly, a community-based screening study in India (*n* = 6120) found that the prevalence of CKD in stages I (7%), II and III (4.3%), and IV and V (0.8%) [21]. Hypertension and diabetes were key risk factors for CKD and hypertension was the commonly occurring co-morbidity among CKD patients (49%) [18,19,20,21,22]. Being hypertensive puts an individual more at-risk for CKD. In India, diabetes and hypertension today account for 40%–60% of cases of CKD [23]. Singh NP et.al., reported that hypertension was associated with low GFR [20]. Even though hypertension is associated with CKD of unknown origin, because of the study nature we cannot establish temporal association. Additionally, studies have found that high blood pressure is common in CKD patients and might have created a biased result.

The worldwide prevalence of CKD is 11%–13% [3,23] with age-standardized prevalence among the 20 years and above population occurring in 10.4% of males and 11.8% of females [24]. In developing countries, the prevalence is 10.6% female and 12.5% male. However, in developed nations 8.6% of males and 9.6% of females are living with CKD [24]. According to a nationally representative sample survey, in China the CKD prevalence among adults is 10.8% [13]. In sub-Saharan Africa, there were poor data quality limits inferences for evidence [10]. The CKD prevalence by stages was stage I (3·5%), stage II (3·9%), stage III (7·6%), stage IV (0·4%), and stage V (0·1%) [3]. Hypertension (48.7%) and diabetes (17.4%) were the most common comorbidities for CKD [25]. The prevalence of end-stage renal failure was about 0.27%. According to MDRD, the prevalence of CKD was around 4% [25] and CKD of undetermined etiology at 16% [17]. 

In recent years, a number of CKD cases without diabetes or hypertension have been reported in several Central American countries, El-Minia Governorate in Egypt, Sri Lanka, particularly in the North Central Province, and Andhra Pradesh, India [2,14,26,27,28]. A community screening in Sri Lanka found that the prevalence of CKD of unknown origin among those above 18 years of age was around 3% [29]. In Sri Lanka, it was found that the prevalence of CKD of unknown origin among males was 13% and females 17%. The risk was higher among the population above 40 years of age [14]. In our findings, the CKD of unknown origin was found more among males (57%), the population below 50 years of age (54%), members of the lower socioeconomic group (70%), and among farmer and agricultural labor (48%). Similarly, in Central America the CKD of unknown origin is found among adult males whose occupation was agriculture, particularly those who were working in sugarcane fields [27]. In our study, groundwater tube-well (49%) and well (41%) water were the main drinking water sources for patients with CKD of unknown origin. The studies revealed that the drinking water might be a risk factor for CKD of unknown origin and the inorganic chemicals in groundwater cause CKD of unknown origin [26,28]. The studies show that patients with CKD of unknown origin are young, poor, and dependent on public hospitals for treatment [17,28,29,30,31], which is similar to our findings.

In many developing countries, CKD is one of the major disease burdens. Hence, it is urgent to understand the epidemiology of CKD [9,10,30,31] and it is important to characterize the responsible factors in order to prevent the disease [12]. In India, the prevention of CKD might be one of the priorities in the planning and implementation of non-communicable disease prevention policy [7]. The management of chronic diseases requires comprehensive, cost-effective, and preventative interventions [11]. There is a critical need for research and interventions to reduce the burdens of CKD in India [5]. The treatment of CKD, especially end-stage renal disease is very expensive. Furthermore, in rural settings management is challenging due to a lack of health care services [31]. There is a lack of trained nephrologists, hence, primary, secondary, and tertiary measures for prevention are significant [29,31] to preventing CKD.

The limitations of this study were that although the target population for screening of CKD was 4133 from 24 sampled villages, 1155 (27.9%) were unwilling to take part in screening because of fear and stigma. Furthermore, this study did not identify etiological factors, as the true prevalence data are lacking and this study focused on the prevalence of CKD.

## 5. Conclusions

This study indicates that early recognition through community screening, awareness-raising among community members, and educating healthcare workers on identifying risk populations is necessary. There is also a need for further research on the etiology of CKD of unknown origin and challenges in healthcare among patients.

## Figures and Tables

**Figure 1 ijerph-17-00456-f001:**
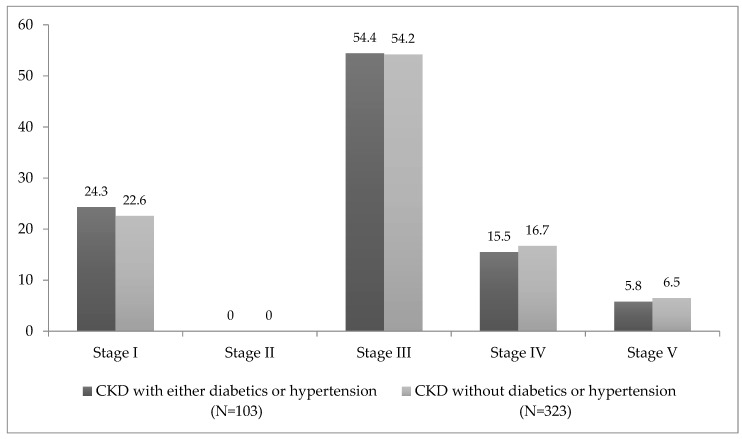
Prevalence of various stages of chronic kidney diseases with either diabetes or hypertension and without diabetes or hypertension.

**Table 1 ijerph-17-00456-t001:** Characteristics of the screening population and prevalence of chronic kidney disease.

Participants’ Characteristics	Screening Population(*N* = 2978)	Total CKD Cases(*N* = 426)	CKD with Either Diabetes or Hypertension(*N* = 103)	CKD without Diabetes or Hypertension(*N* = 323)
*n*	%	*n*	%	*n*	%	*P*	*n*	%	*P*
Sex							0.044			<0.001
Male	1112	37.3	231	54	48	47		183	57	
Female	1866	62.7	195	46	55	53		140	43	
Age in years							0.001			<0.001
20–30	564	18.9	22	5	1	1		21	6	
31–40	810	27.2	60	14	9	9		51	16	
41–50	852	28.6	138	33	34	33		104	32	
51–60	752	25.3	206	48	59	57		147	46	
Education							0.683			<0.001
Illiterate	532	17.9	107	25	24	23		83	26	
Primary school (1–5)	983	33.0	170	40	34	33		136	42	
Upper primary school (6 and 7)	402	13.5	45	10	14	14		31	10	
High school (8–10)	910	30.6	88	21	27	26		61	19	
Intermediate (11 and 12)	75	2.5	9	2	1	1		8	2	
Graduate (13 and above)	76	2.5	7	2	3	3		4	1	
Marital status							0.264			<0.001
Unmarried	187	6.3	7	2	1	1		6	2	
Married	2662	89.4	384	90	93	90		291	90	
Widowed	129	4.3	35	8	9	9		26	8	
Caste							0.286			<0.001
General	608	20.4	96	22	29	28		67	21	
Other Backward Class (OBC)	1696	56.9	203	48	55	53		148	46	
Scheduled Caste (SC)	601	20.2	107	25	17	17		90	28	
Scheduled Tribe (ST)	73	2.5	20	5	2	2		18	5	
Sub-caste							0.354			<0.001
Badhei/Kumbhar	167	5.6	24	6	7	7		17	5	
Bania/Teli/Rangani	356	11.9	38	9	15	14		23	7	
Bhandari	95	3.2	7	2	1	1		6	2	
Brahmin/Mali	101	3.4	15	4	5	5		10	3	
Chamar/Kandar/Pana	235	7.9	37	9	7	7		30	9	
Chasa/sudra	621	20.8	79	18	20	19		59	18	
Dhoba	28	0.9	5	1	1	1		4	1	
Gopal	318	10.7	38	9	6	6		31	10	
Gudia	68	2.3	11	3	5	5		6	2	
Kaibarta	315	10.6	55	13	8	8		47	15	
Kandha/Khaira	73	2.4	20	4	2	2		18	6	
Karana	36	1.2	3	1	3	3		0	0	
Khandayat	513	17.2	82	19	21	21		61	19	
Tainla/Taira	49	1.6	12	3	1	1		11	3	
Gotra							0.131			0.062
Bachchas	101	3.4	19	4	0	0		19	6	
Baghasya	65	2.2	9	2	1	1		8	2	
Bharadwaja	51	1.7	9	2	4	4		5	2	
Gajasya	186	6.2	10	3	3	3		7	2	
Kachhapa	113	3.8	11	3	5	5		6	2	
Nagasya	2013	68.0	299	70	69	67		230	71	
Other	449	14.7	69	16	21	20		48	15	
Occupation							0.065			<0.001
Agriculture	416	14.0	105	25	21	20		84	26	
Government services	75	2.5	16	4	7	7		9	3	
Private services	51	1.7	2	0.4	0	0		2	0.6	
Daily labor	489	16.4	85	20	15	15		70	22	
Business	134	4.5	31	7	8	8		23	7	
Homemakers	1756	59.0	185	43	52	50		133	41	
Student	57	1.9	2	0.6	0	0		2	0.4	
Socioeconomic status							0.590			0.007
Lower	1816	61.0	284	67	56	54		228	70	
Middle	968	32.5	120	28	37	36		83	26	
Upper	194	6.5	22	5	10	10		12	4	
Main source of drinking water							0.786			0.176
Tube-well	1305	43.8	203	48	44	43		159	49	
Well	1388	46.6	186	43.6	52	50		134	41	
Supply water	276	9.3	35	8	7	7		28	9	
River	9	0.3	2	0.4	0	0		2	1	
Alcohol consumption							0.276			<0.001
Never	2543	85.4	335	79	89	86		246	76	
Desi liquor	23	0.8	9	2	2	2		7	2	
Foreign liquor	257	8.6	44	10	5	5		39	12	
Both desi and foreign liquor	155	5.2	38	9	7	7		31	10	
Smoking or chewing tobacco							0.179			0.076
Never	2072	69.6	221	52	63	61		158	49	
Occasionally smoke	88	2.9	16	4	1	1		15	4	
Regularly smoke	175	5.9	50	12	9	9		41	13	
Smokeless tobacco	644	21.6	139	32	30	29		109	34	
Body Mass Index (BMI)							0.758			0.066
<18.5	201	6.8	39	9	5	5		34	10	
18.5 to 25	2381	80.0	337	79	80	78		257	80	
26 to 30	332	11.1	39	9	11	10		28	9	
>30	64	2.1	11	3	7	7		4	1	
Family members diagnosed with CKD							<0.001			<0.001
Yes	203	6.8	84	20	21	20		63	19	
No	2775	93.2	342	80	82	80		260	81

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
