# Peer review of "Prevalence of Chronic Kidney Disease in Cuttack District of Odisha, India"

_ijerph, 2020, doi:10.3390/ijerph17020456_

Round 1
Reviewer 1 Report
Thank you for submitting your paper on the prevalence of CKDu in Narsinghpur, India.
Although the topic is of utmost interest your paper has a number of issues the need to be addressed. Your report lacks details on how diabetes and hypertension was defined. No reference is provided for the definition of CKD. The description of the statistical analysis is missing. The mean serum creatinine and eGFR is also missing.
There is no information on the population without CKD and how they compare with those with CKD. A sound statistical analysis, including a multivariate regression analysis to identify risk factors for CKD, is missing.
In table 1 the total CKD population with either hypertension or diabetes is 426 (14.3%). However in table 2 that number corresponds to the total population with CKD. Table 1 is too crowded with data that could easily be described in the text.
The manuscript has multiple gramatical errors.
Author Response
Response to Reviewers
Dear editor,
We thank the two reviewers for their comments. The constructive comments/ suggestions by the reviewer are really appreciated. We have now completely revised the manuscript.
Please find enclosed our revised manuscript. We have revised the manuscript accordingly and provide specific answers below. In the following, we respond to the individual remarks; Reviewer’s comments are identified by RC and authors’ by AC.
Reviewer 1
RC: Thank you for submitting your paper on the prevalence of CKDu in Narsinghpur, India. Although the topic is of utmost interest your paper has a number of issues the need to be addressed.
AC: As per the suggestions the background of the study revised and all relevant references included. Similarly, we revised the methods, results, and conclusions. We have done thorough English editing and corrected the grammatical mistakes in the revised manuscript.
RC: Your report lacks details on how diabetes and hypertension were defined.
AC: The definition of diabetes and hypertension was provided on page 2 in section 2.3.
RC: No reference is provided for the definition of CKD.
AC: Definition and reference of CKD are given on page 1 paragraph 1.
RC: The description of the statistical analysis is missing.
AC: The detail statistical analysis section is provided in section 2.4.
RC: The mean serum creatinine and eGFR is also missing.
AC: Mean serum creatinine and eGFR value are provided on page 3 Results section.
RC: There is no information on the population without CKD and how they compare with those with CKD. A sound statistical analysis, including a multivariate regression analysis to identify risk factors for CKD, is missing.
AC: Thanks for the suggestion. Although CKD is the severe NCD in the study setting, the true prevalence data are lacking; hence, the study focused on the prevalence of CKD. However, our second manuscript is a case-control study, which aims to find out the etiology, where we will include multivariate regression analysis to identify risk factors for CKD.
RC: In table 1 the total CKD population with either hypertension or diabetes is 426 (14.3%). However, in table 2 that number corresponds to the total population with CKD. Table 1 is too crowded with data that could easily be described in the text.
AC: As per the suggestion ‘Table 1. Details of screening population and village-specific information on chronic kidney diseases with either diabetics or hypertension and without diabetics or hypertension.’ removed and data described in the text.
RC: The manuscript has multiple grammatical errors.
AC: We have done thorough English editing and corrected the grammatical mistakes in the revised manuscript.
Reviewer 2 Report
The present study by Mohanty et al examined the prevalence of (unknown) origin of chronic kidney disease in Narsinghpure, a block area in Odisha, India. This is a cross-sectional study which included about 2978 patients from 20-60 years and found 14% diagnosed as CKD and 10% with unknown origin of CKD. Author is appreciated to examined under studied Indian population on CKD where much of the epidemiological and centralized CKD data base are missing compared to western counterparts. Overall, it is welcomed epidemiological study and the manuscript has intrinsic value with of course, much of caveats.
Comments;
Author claims to be diabetes and hypertension is known causes (origin) of CKD while causes other than these two are unknown causes. It is true that diabetes and hypertension are main causes of CKD which are responsible for up to two-thirds of the CKD cases. Both of these causes leads to an end stage organ damage. Polycystic kidney disease and Lupus another know cause of CKD which is highly prevalent among Indian women. Glomerulonephritis is also one of the main cause of CKD. There are other minor causes such as obstructions caused by problems like kidney stones, tumors or an enlarged prostate gland in men and, repeated urinary infections. Authors have never mentioned or addressed about above mentioned causes. When already known causes has not been addressed in these patient, it is not wise to call an unknow origin that even undermine the available literature on CKD. Author should include inclusion and exclusion criteria with box and diagram. Please mention all catalog number for kits that are used in this study. Line 74: what is urine blood? Please explain or correct the text. Line 59: Include reference for Varma et al. Line 82: Include reference for guideline. Table: Separate total CKD with diabetes and hypertension and third column together to distinguish better. Discussion part mainly discussing about existing work. Author should discuss more of the current work and rationalizing with previously existing literature. Author should include more recent works for the reference. Limitation of the study should be included.a. Author drawn attention as backward classes and scheduled castes has higher prevalence of CKDu. Please speculate why do they have such a condition.
b. Study is about 20-60 years. Does this condition arose recently? Or is there food, cement, agriculture factories in and around Narsinghpure that might be causing upraising CKD.
c. Apparently ground water tubewell (49%) drinking water causes more CKDu than well water (41%)
Author Response
Response to Reviewers
Dear editor,
We thank the two reviewers for their comments. The constructive comments/ suggestions by the reviewer are really appreciated. We have now completely revised the manuscript.
Please find enclosed our revised manuscript. We have revised the manuscript accordingly and provide specific answers below. In the following, we respond to the individual remarks; Reviewer’s comments are identified by RC and authors’ by AC.
Reviewer 2
RC: The present study by Mohanty et al examined the prevalence of (unknown) origin of chronic kidney disease in Narsinghpure, a block area in Odisha, India. This is a cross-sectional study which included about 2978 patients from 20-60 years and found 14% diagnosed as CKD and 10% with unknown origin of CKD. Author is appreciated to examined under studied Indian population on CKD where much of the epidemiological and centralized CKD data base are missing compared to western counterparts. Overall, it is welcomed epidemiological study and the manuscript has intrinsic value with of course, much of caveats.
AC: Thanks for the appreciation. As per the suggestions the background of the study revised and all relevant references included. Similarly, we revised the methods, results, and conclusions. We have done thorough English editing and corrected the grammatical mistakes in the revised manuscript.
RC: Author claims to be diabetes and hypertension is known causes (origin) of CKD while causes other than these two are unknown causes. It is true that diabetes and hypertension are main causes of CKD which are responsible for up to two-thirds of the CKD cases. Both of these causes leads to an end stage organ damage. Polycystic kidney disease and Lupus another know cause of CKD which is highly prevalent among Indian women. Glomerulonephritis is also one of the main cause of CKD. There are other minor causes such as obstructions caused by problems like kidney stones, tumors or an enlarged prostate gland in men and, repeated urinary infections. Authors have never mentioned or addressed about above mentioned causes. When already known causes has not been addressed in these patient, it is not wise to call an unknow origin that even undermine the available literature on CKD.
AC: As this study is not addressed other known causes (origin) of CKD except diabetes and hypertension; we removed ‘unknow origin’ from the title and objective.
RC: Author should include inclusion and exclusion criteria with box and diagram.
AC: The inclusion and exclusion criteria of the study participants were provided in the page 2 method section.
RC: Please mention all catalog number for kits that are used in this study.
AC: The catalog number is given in section 2.3.
RC: Line 74: what is urine blood? Please explain or correct the text.
AC: Correction has been made in the revised manuscript.
RC: Line 59: Include reference for Varma et al.
AC: Included the reference for Varma et al.
RC: Line 82: Include reference for guideline.
AC: Included the reference for guideline Ref 19.
RC: Table: Separate total CKD with diabetes and hypertension and third column together to distinguish better.
AC: As only 18 individuals were having CKD with both diabetics and hypertension, these findings were not presented in Table. It is provided in text in Page 5 paragraph 1.
RC: Discussion part mainly discussing about existing work. Author should discuss more of the current work and rationalizing with previously existing literature. Author should include more recent works for the reference.
AC: We revised the discussion as per the suggestion.
RC: Limitation of the study should be included.
AC: A new paragraph on limitations was included in the discussion section
RC: a. Author drawn attention as backward classes and scheduled castes has higher prevalence of CKDu. Please speculate why do they have such a condition. b.Study is about 20-60 years. Does this condition arose recently? Or is there food, cement, agriculture factories in and around Narsinghpure that might be causing upraising CKD. c. Apparently ground water tubewell (49%) drinking water causes more CKDu than well water (41%)
AC: AC: Thanks for the suggestion. Although CKD is the severe NCD in the study setting, the true prevalence data are lacking; hence, the study focused on the prevalence of CKD. However, our second manuscript is a case-control study, which aims to find out the etiology, where we will include multivariate regression analysis to identify risk factors for CKD. In this all possible exposure will be included.
Round 2
Reviewer 2 Report
Authors were addressed all the questions raised.
This manuscript is a resubmission of an earlier submission. The following is a list of the peer review reports and author responses from that submission.